# *Candida lipolytica* Bloodstream Infection in an Adult Patient with COVID-19 and Alcohol Use Disorder: A Unique Case and a Systematic Review of the Literature

**DOI:** 10.3390/antibiotics12040691

**Published:** 2023-04-01

**Authors:** Omar Simonetti, Verena Zerbato, Sara Sincovich, Lavinia Cosimi, Francesca Zorat, Venera Costantino, Manuela Di Santolo, Marina Busetti, Stefano Di Bella, Luigi Principe, Roberto Luzzati

**Affiliations:** 1Infectious Diseases Unit, Trieste University Hospital (ASUGI), 34125 Trieste, Italy; 2Operative Unit of Medicina Clinica, Trieste University Hospital (ASUGI), 34125 Trieste, Italy; 3Microbiology Unit, Trieste University Hospital (ASUGI), 34125 Trieste, Italy; 4Clinical Department of Medical, Surgical and Health Sciences, Trieste University, 34149 Trieste, Italy; 5Clinical Pathology and Microbiology Unit, “S. Giovanni di Dio” Hospital, 88900 Crotone, Italy

**Keywords:** *Candida lipolytica*, *Yarrowia lipolytica*, fungemia, candidemia, alcohol use disorder, COVID-19, caspofungin, fluconazole

## Abstract

*Candida lipolytica* is an uncommon *Candida* species causing invasive fungemia. This yeast is mainly associated with the colonisation of intravascular catheters, complicated intra-abdominal infections, and infections in the paediatric population. Here, we report a case of *C. lipolytica* bloodstream infection in a 53-year-old man. He was admitted for an alcohol withdrawal syndrome and mild COVID-19. Among the primary risk factors for candidemia, only the use of broad-spectrum antimicrobials was reported. The empiric treatment was commenced with caspofungin and then targeted with intravenous fluconazole. Infective endocarditis was ruled out using echocardiography, and PET/TC was negative for other deep-seated foci of fungal infection. The patient was discharged after blood culture clearance and clinical healing. To the best of our knowledge, this is the first case of *C. lipolytica* candidemia in a patient with COVID-19 and alcohol use disorder. We performed a systematic review of bloodstream infections caused by *C. lipolytica*. Clinicians should be aware of the possibility of *C. lipolytica* bloodstream infections in patients with alcohol use disorder, especially in a COVID-19 setting.

## 1. Introduction

*Candida* species are common pathogens causing nosocomial bloodstream infections (BSIs) worldwide and account for more than 90% of fungal BSIs [1]. The incidence of nosocomial *Candida* BSIs is quite variable, ranging between 0.3 and 5 per 1000 admissions [2]. There are several well-known risk factors for candidemia, including central venous catheterisation, parenteral hyperalimentation, broad-spectrum antibiotics, intensive care, malignancies, haematological aberrations, and immunocompromised conditions [3].

Historically, *Candida albicans* was the major pathogen of candidemia; nevertheless, in recent years, non-*albicans Candida* species have been responsible for up to 50% of all candidemia cases in some settings [4].

The incidence of *Candida* spp. BSIs in patients with COVID-19 syndrome is significantly higher than in patients without this syndrome. Various explanations have been suggested, such as poor central venous catheter (CVC) care, immunosuppression (e.g., the administration of tocilizumab or high doses of corticosteroids), and the increased use of antibiotics and their effects on the gut microbiome [5]. Furthermore, both viral infection and microthrombi formation can alter the gut–blood barrier, resulting in intestinal microbiota entering the blood [6].

*Candida lipolytica* (*Yarrowia lipolytica*) is a ubiquitous ascomycetous yeast growing in the environment, meat, and cheese products. It can occasionally colonise the gut and faeces, oropharynx, and the skin of asymptomatic persons [7]. *C. lipolytica* infection in humans was firstly reported by Wehrspann and Füllbrandt in 1985 [8]. Although this fungus was previously considered a low-virulence yeast, increasing episodes of nosocomial infections (i.e., catheter-related BSIs) in immunocompromised or critically ill patients have been recently reported. One hypothesis about the invasiveness of *C. lipolytica*, in addition to the production of proteases and lipases, is the ability to form biofilms [7]. Invasive *C. lipolytica* infections, except for catheter-related *C. lipolytica* infections, occurred in the context of traumatic ocular infection, an acute exacerbation of chronic sinusitis and acute pancreatitis [3,7]. Here, we present a unique case of *C. lipolytica* BSI in a patient affected by alcohol use disorder and concomitant COVID-19. Furthermore, we performed a systematic literature review of candidemia episodes due to *C. lipolytica*.

## 2. Materials and Methods

This systematic review was performed according to the Preferred Reporting Items for Systematic reviews and Meta-Analyses (PRISMA) [9]. This systematic review is registered with PROSPERO (CRD42023405326).

The PubMed database was searched for articles published from inception until 15th December 2022 using the following combination of keywords: “Candida lipolytica”[Title/abstract] OR “Yarrowia lipolytica”[Title/abstract]) AND (“candidemia”[Title/abstract] OR “infection*”[Title/abstract] OR “fungaemia”[Title/abstract] OR “fungemia”[Title/abstract] OR “case*”[Title/abstract] OR “candidiasis”[Title/abstract] OR “fungus disease*”[Title/abstract] OR “fungus infection*”[Title/abstract] OR “fungal infection*”[Title/abstract] OR “fungal disease*”[Title/abstract] OR “bloodstream infection*”[Title/abstract] OR “BSI”[Title/abstract].

One investigator (V.Z.) carried out the first selection of retrieved records by screening their titles and abstracts in order to establish eligibility for a full-text review. The second step (performed by V.Z., S.S. and L.C.) consisted of the further screening of full-text articles to define final inclusion in the systematic review according to the inclusion criteria. We included full texts (written in English, Spanish and German) of case reports, case series and systematic reviews about *Candida lipolytica* BSI. We excluded papers containing only microbiological data (e.g., susceptibility and genomics). Additional cases were sought from the reference list of included papers and reviews.

The following information was extracted from each article and entered into pilot-tested evidence tables: author, year, the country of diagnosis, age, gender, the origin of infection, underlying conditions and risk factors (immunodeficiency, parenteral nutrition and previous abdominal surgery), source control, clinical presentation (septic shock, coinfections and complications), susceptibility to main antifungals, antifungal therapy and outcomes.

## 3. Case Presentation

On 3 June 2022, a 53-year-old man entered the emergency room for alcohol withdrawal syndrome associated with chronic alcoholism. He was a regular smoker, and his past medical history was unremarkable, except for reported episodes of unspecified haematemesis. He was further hospitalised for acute care because of withdrawal symptoms. 

On admission to a medical ward, he was found to have a right submandibular tumour. Routine laboratory tests were unremarkable, and a screening nasal swab was negative for SARS-CoV-2. A computed tomography (CT) scan of the neck and head revealed a productive submandibular lesion with no homogeneous enhancement. The patient underwent an ultrasound-guided needle biopsy of the right parotid gland, and its histology was consistent with possible sialocele or adenoma.

The above-mentioned CT examination also described hypodense material in all mastoid cells and in the right tympanic cavity with a retracted tympanic membrane, as well as a suspected focal brain lesion. Magnetic resonance imaging (MRI) of the brain demonstrated both cholesteatomatous otitis in the right tympanic cavity and a cerebral cavernoma. The patient received empirical antibiotic therapy with piperacillin/tazobactam from 13 to 22 June. On 30 June, the patient was found to be positive for SARS-CoV-2 at a screening using a nasal swab and then transferred to the COVID-19 department. Then, mild febrile COVID-19 was diagnosed, and another course of piperacillin/tazobactam was prescribed.

Blood cultures were found to be positive for *C. lipolytica* using Matrix-Assisted Laser Desorption Ionization Time-of-Flight (MALDI-TOF) technology (Figure 1). At that time and following patient admission, no CVC was present. The abnormal blood test results were as follows: C-reactive protein 98.1 mg/L (reference value < 5 mg/L) and procalcitonin 0.73 μg/L (reference value < 0.5 μg/L). Caspofungin was started pending sensitivity tests and the BSI clearance. A transthoracic echocardiography was negative for endocarditis, and a funduscopic examination ruled out retinal embolisms. In addition, positron emission tomography (PET/CT) ruled out other deep-seated foci of fungal infection. 

After 72 h of echinocandin treatment, the patient was afebrile, and three blood culture sets were negative. Thus, the patient received caspofungin up to 15 July and subsequently concluded a total of 14 days of targeted therapy with fluconazole since bloodstream clearance. 

In summary, we report a BSI episode due to *C. lipolytica* of unknown origin in a patient with alcohol use disorder and concomitant mild COVID-19.

## 4. Review

The literature search identified 17 articles about cases of *Candida lipolytica* bloodstream infections (Figure 2).

We found 89 cases of *C. lipolytica* BSI (Table 1). The first case was described by Wehrspann and Fullbrandt in 1985 in Germany [8].

The reported candidemias were described mostly in Asia and particularly in the following countries: China (*n* = 14) [7,14], Korea (*n* = 6) [16,20], Taiwan (*n* = 3) [3,13,19], Turkey (*n* = 3) [12,18], India (*n* = 1) [17] and Qatar (*n* = 1) [21].

The other cases were reported in Africa (Tunisia, *n* = 55, in the context of an outbreak) [23], Europe (Spain *n* = 3; Italy *n* = 1; Germany *n* = 1) [8,10,11,15] and the United States (*n* = 1) [22].

The mean age of the affected patients was 41.48 years old (+/−23.86). A total of 76% of the patients were males (*n* = 68). Candidemia was catheter-related in 94% of cases (84 out of 89 subjects). Regarding other predisposing risk factors, parenteral nutrition was described in 25 patients (28%), while previous abdominal surgery and immunodeficiency were described in 20 and 14 patients (23% and 16%), respectively. 

Candidemia occurred with septic shock in 16 patients (18%). Sixty-nine patients (78%) received antifungal therapy. In 15 cases (17%), antifungal combination therapy was prescribed. Of these, 87% of patients received targeted therapy with amphotericin B and fluconazole, while 13% received amphotericin B and caspofungin [12,23]. In total, 5 out of 15 (67 %) of the patients who received antifungal combination therapy died. Amphotericin B and fluconazole were the most prescribed drugs. The resolution of the candidemia was reported in 53 patients (60%). For 11 surviving patients, the treatment only consisted of source control with CVC removal. Death occurred in 34 cases (38%). Source control was not carried out in almost one-third of the patients who died (12 patients out of 34).

## 5. Discussion

The majority of patients with *C. lipolytica* BSI described in the literature are represented by adult males, characteristics consistent with our experience. Furthermore, the presentation of symptoms described herein was mild, in line with previous reports where septic shock was rare. Interestingly, the case we described was a *C. lipolytica* candidemia not related to the presence of an intravascular device. As a matter of fact, the source of the candidemia was never detected. This is in contrast with the results of our literature review, which shows a high burden of *C. lipolytica* fungemia related to CVC. This yeast is ubiquitous both in hospital environments and at the community level. Nevertheless, because of its selective ability to adhere and form a biofilm on medical devices, the majority of invasive infections encountered are of nosocomial origin [7,24].

Our case is a unique report of *C. lipolytica* BSI in a patient suffering from COVID-19 without indwelling catheters as a risk factor for candidemia; however, the patient had a history of alcohol use disorder (AUD). Such behaviour profoundly alters the gut microbiome, increases intestinal permeability, causes gut dysfunction, induces bacterial translocation and exacerbates the process of alcohol-associated liver disease (ALD). Furthermore, ethanol abuse decreases the prevalence of *Epicoccum*, *Galactomyces* and *Debaryomyces* in the gut, while *Candida* spp. burden increases significantly [25]. Indeed, AUD has been previously described to be a risk factor for *C. lipolytica* BSI, as shown by two reports included in our analysis [8,22].

In addition, our patient underwent two courses of therapy with a broad-spectrum antibiotic (piperacillin/tazobactam). Antibacterial drugs have a long-term effect on the microbiome of the human gut by shifting fungal communities from mutualism to competition and reducing the abundance of bacteria that actively suppress the pathogenicity of opportunistic fungi, such as *Candida* spp. Nevertheless, piperacillin/tazobactam does not seem to be associated with invasive fungal infections like other classes of antibiotics, such as fluoroquinolones [26,27]. 

The risk factors and incidence of invasive candidiasis in patients with COVID-19 are in the progress of being defined. Indeed, the frequency of this fungal infection ranges from 0.03 to 14% because of the heterogeneity of patients and cohorts of study [28]. The faecal microbiome was also studied in patients hospitalised for COVID-19, and an intestinal accumulation of fungal pathogens belonging to the genera *Candida* and *Aspergillus* was found compared to controls [29]. Other authors showed that SARS-CoV-2 intestinal mucosa damage and malnutrition correlate with secondary infection, such as bloodstream infections [6]. Thus, our patient, who appeared to have no primary risk factor for invasive candidemia, had really three underlying conditions favouring intestinal *C. lipolytica* translocation, namely, AUD, COVID-19, and broad-spectrum antibiotic therapy. 

There is a scarcity of data on *C. lipolytica* BSI outcomes. Our analysis revealed a high mortality rate in affected patients. Nevertheless, up to 12% of the included patients survived without any antimycotic therapy. As the management of this rare infection is not standardised, we decided to follow the available guidelines and prescribe a minimum duration of therapy of 2 weeks after the documented clearance of *Candida* from the bloodstream [1]. International guidelines for the treatment of candidiasis generally do not include combination therapy, except in certain clinical cases, such as endocarditis. Nevertheless, some combination therapies seem to have a synergistic effect against difficult-to-treat *Candida* species by preventing or reducing biofilm formation [30]. Although *C. albicans* remains the most pathogenic yeast, the selective abilities of *C. lipolytica* to form biofilms on devices and produce haemolytic enzymes are of particular interest in a nosocomial setting [24]. The role of antimycotic combination therapy needs to be further studied. The patient here described recovered after prolonged antifungal monotherapy and was discharged from our institution in good clinical condition. 

## 6. Conclusions

We described a unique case of *C. lipolytica* BSI in a patient with AUD, COVID-19, and antibiotic therapy, which all represent conditions favouring the intestinal translocation of *Candida* spp. We also conducted a systematic review of previously published cases of *C. lipolytica* candidemia. This study reinforces the available data on the specific risk factors for such an invasive fungal infection and contributes with a personal perspective on its management.

## Figures and Tables

**Figure 1 antibiotics-12-00691-f001:**
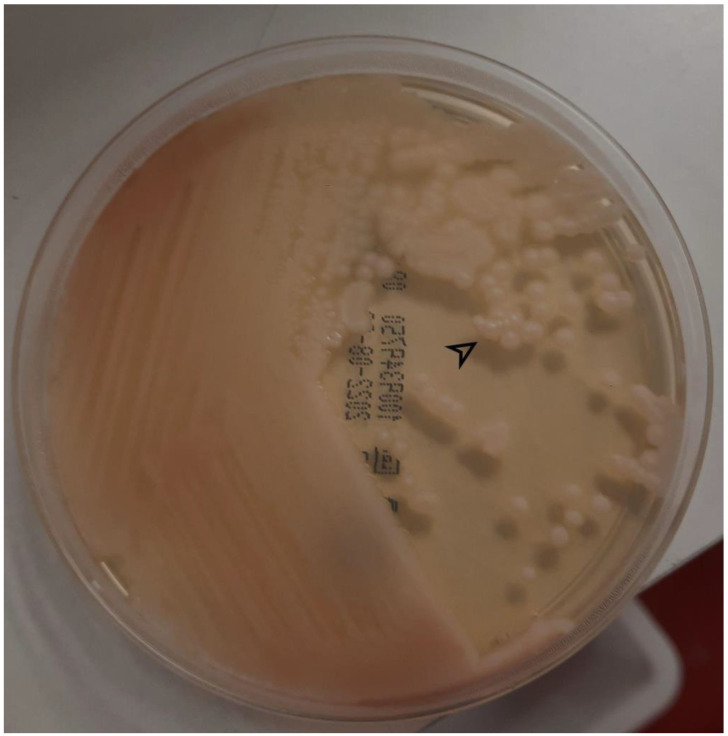
*C. lipolytica* colonies (arrow) growing on Sabouraud agar.

**Figure 2 antibiotics-12-00691-f002:**
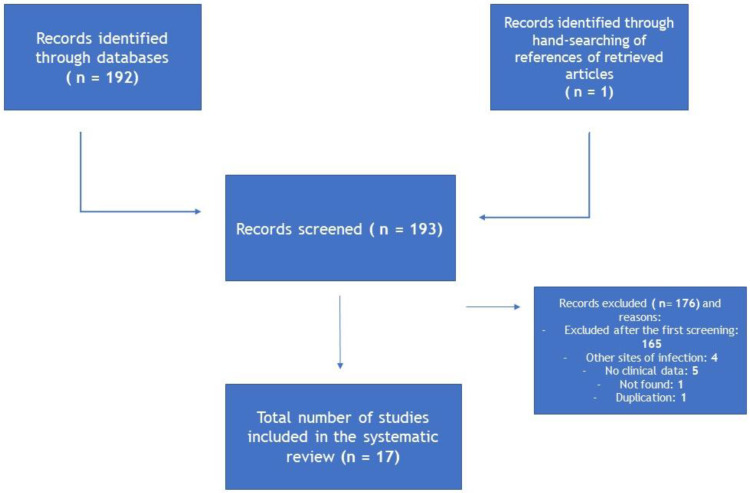
Literature search strategy.

**Table 1 antibiotics-12-00691-t001:** Cases of *C. lipolytica* BSI.

Author/ Year	Country	Age^ Gender	Catheter-Related	Immunocompromising Underlying Disease	PN	Recent Abdominal Surgery *	Underlying Diseases	Source Control (Y/N)	Septic Shock (Y/N)	Treatment (Drug and Duration)	Complications (Persistence, Recurrence, Other)	Coinfections	Outcome	Ref.
Marín Martínez, 2016	Spain	86 F	Y	N	N	Y	DM	NR	N	NR	N	N	NR	[10]
Blanco, 2009	Spain	12 F	Y	Y (pancreatic fibrosis)	Y	N	Cystic pancreatic fibrosis	Y	N	Voriconazole (MIC 0.06 mg/L)	N	Pneumonia	Resolution	[11]
Blanco, 2009	Spain	86 F	Y	Y (bladder cancer)	N	N	Hypertension; DM; dementia; UTI	Y	N	Caspofungin (MIC 0.5 mg/L)	N	N	Resolution	[11]
Belet, 2015	Turkey	2 days, M	Y	N	Y	Y	Intestinal obstruction	Y	N	AMB (24 d) + caspofungin (10 d)	Persistence	N	Resolution	[12]
Belet, 2015	Turkey	4 months, F	Y	N	Y	N	Gastroesophageal reflux	Y	N	Amphotericin B + caspofungin	Persistence	MRSA pneumonia	Resolution	[12]
Chi, 2017	Taiwan	44 M	Y	Y (gastric cancer)	N	Y	Gastric cancer	N	Y	Voriconazole (MIC 0.06 μg/mL)	N	*A. baumannii* BSI, *C. tropicalis* BSI (in therapy)	Death	[13]
Ye, 2011	China	13 M	Y	Y (ALL)	N	N	ALL	Y	N	Fluco	N	N	Resolution	[14]
D’Antonio, 2002	Italy	18 F	Y	Y (allogeneic BMT)	N	N	ALL	Y	N	AMB (1st episode: 10 d, MIC 0.39 μg/mL)	Recurrence	CMV pneumonia	Death	[15]
Chang, 2001	Korea	15 F	Y	Y (AML)	N	N	AML	Y	N	None	Recurrence	*S. malthophilia* BSI	Resolution	[16]
Agarwal, 2008	India	2 M	Y	N	N	N	N	Y	N	AMB (2nd episode)	Recurrence	TB meningitis	Resolution	[17]
Ozdemir, 2011	Turkey	9 M	Y	Y (neuroblastoma)	Y	N	Relapsed neuroblastoma	N	N	Caspofungin (14 d after fungemia clearance) + lock	N	N	Resolution	[18]
Lai, 2012	Taiwan	61 M	Y	Y (lung cancer)	N	N	Lung cancer	Y	Y	Fluco (MIC 1 μg/mL), then micafungin	Persistence	*C. albicans* BSI, pneumonia	Resolution	[19]
Shin, 2000	Korea	1 month, M	Y	N	Y	N	Necrotisingenterocolitis	Y	N	Fluco (6 d, MIC 32 μg/mL)	Persistence	N	Resolution	[20]
Shin, 2000	Korea	2 month, M	N	N	N	N	N	N	N	None	N	Streptococcal meningitis	Resolution	[20]
Shin, 2000	Korea	8 F	Y	Y (AML)	N	N	AML	N	N	AMB (6 d, MIC 0.5 μg/mL)	Persistence	N	Resolution	[20]
Shin, 2000	Korea	14 M	Y	Y (AML)	Y	N	AML	N	N	AMB (21 d, MIC 0.5 μg/mL)	Persistence	N	Resolution	[20]
Shin, 2000	Korea	4 M	Y	Y (aplastic anaemia)	N	N	Aplastic anaemia	Y	N	Fluco (MIC 32 μg/mL)	Persistence	N	Resolution	[20]
Taj-Aldeen, 2014	Qatar	77 F	NR	N	N	N	DM, renal failure	NR	NR	Caspofungin (MIC 2 μg/mL)	NR	NR	Death	[21]
Walsh, 1989	USA	54 M	Y	N	N	Y	Alcohol abuse, cholelithiasis	Y	N	None	Catheter-related thrombophlebitis, persistence	N	Resolution	[22]
Wehrspann, 1984	Germany	57 F	Y	Y	N	N	Alcohol abuse, stroke, peptic ulcer	Y	N	Ketoconazole	N	Infective endocarditis	Resolution	[8]
Liu, 2013	Taiwan	84 M	N	N	N	N	Acute pancreatitis	N	N	Fluco (MIC 1 μg/mL), then micafungin (total 14 d)	Persistence	N	Resolution	[3]
Zhao, 2015	China	65 M	Y	N	NR	Y	Polytrauma with subdural haematoma	Y	NR	Fluco (MIC 4 μg/mL)	N	N	Resolution	[7]
Zhao, 2015	China	46 M	Y	N	NR	N	Rheumatic heart disease	Y	NR	None	N	N	Resolution	[7]
Zhao, 2015	China	27 M	Y	N	NR	N	Cervical spinal fracture, paraplegia	Y	NR	None	N	N	Resolution	[7]
Zhao, 2015	China	67 M	Y	N	NR	N	Stroke	Y	NR	Itraconazole (MIC 0.5 μg/mL)	N	N	Resolution	[7]
Zhao, 2015	China	73 M	Y	Y (pancreatic cancer)	NR	N	Pancreatic cancer	Y	NR	AMB (MIC 0.5 μg/mL)	N	Recurrent peritonitis	Resolution	[7]
Zhao, 2015	China	1 M	Y	N	NR	N	Prematurity	Y	NR	None	N	N	Resolution	[7]
Zhao, 2015	China	3 M	Y	N	NR	N	Congenital heart disease	Y	NR	Fluco (MIC 16 μg/mL)	N	N	Resolution	[7]
Zhao, 2015	China	63 M	Y	N	NR	N	Bronchiectasis	Y	NR	Fluco (MIC > 256 μg/mL)	N	N	Death	[7]
Zhao, 2015	China	75 M	Y	N	NR	N	Cerebral haemorrhage	Y	NR	Fluco (MIC 64 μg/mL)	N	N	Death	[7]
Zhao, 2015	China	43 M	Y	N	NR	N	Brainstem death	Y	NR	None	N	Pneumonia	Resolution	[7]
Zhao, 2015	China	45 M	Y	N	NR	Y	N	Y	NR	Fluco (MIC 128 μg/mL)	N	N	Resolution	[7]
Zhao, 2015	China	73 M	Y	N	NR	N	N	Y	NR	Fluco (MIC 64 μg/mL)	N	N	Resolution	[7]
Zhao, 2015	China	82 F	Y	N	NR	N	N	Y	NR	Fluco (MIC 64 μg/mL)	N	N	Death	[7]
Trabelsi **, 2015	Tunisia	21 M	Y	N	Y	Y	Polytraumatism	Y	N	AMB (13 d) + Fluco (17 d)	N	N	Resolution	[23]
Trabelsi, 2015	Tunisia	39 M	Y	N	N	N	Polytraumatism	Y	N	None	N	N	Resolution	[23]
Trabelsi, 2015	Tunisia	60 F	Y	N	Y	N	DM, stroke	Y	Y	Fluco (3 d) + AMB (6 d)	N	Pneumonia	Death	[23]
Trabelsi, 2015	Tunisia	48 F	Y	N	N	N	Polytraumatism	Y	N	Fluco (7 d)	N	Pneumonia	Resolution	[23]
Trabelsi, 2015	Tunisia	32 F	Y	N	N	N	DM	N	N	Fluco (3 d) + AMB (1 d)	N	N	Death	[23]
Trabelsi, 2015	Tunisia	15 F	Y	N	N	Y	Acute anaemia	Y	N	Fluco (9 d)	N	N	Death	[23]
Trabelsi, 2015	Tunisia	78 M	Y	N	N	N	COPD	N	N	Fluco (7 d)	N	N	Resolution	[23]
Trabelsi, 2015	Tunisia	50 M	Y	Y (colon cancer)	Y	Y	Colon cancer	N	N	Fluco (6 d) + AMB (17 d)	N	Peritonitis	Death	[23]
Trabelsi, 2015	Tunisia	60 M	Y	N	N	N	CKD, DM	N	Y	AMB (2 d)	N	N	Death	[23]
Trabelsi, 2015	Tunisia	47 M	Y	N	N	Y	Polytraumatism	Y	N	AMB (10 d)	N	Pneumonia	Resolution	[23]
Trabelsi, 2015	Tunisia	27 M	Y	N	N	Y	Thoracic traumatism	Y	N	Fluco (23 d)	N	N	Resolution	[23]
Trabelsi, 2015	Tunisia	43 M	Y	N	N	N	Polytraumatism	Y	N	Fluco	N	N	Resolution	[23]
Trabelsi, 2015	Tunisia	52 M	Y	N	N	N	Renal failure, stroke	Y	N	Fluco (2 d) + AMB (7 d)	N	Pneumonia	Death	[23]
Trabelsi, 2015	Tunisia	68 M	N	N	N	N	COPD	N	N	Fluco	N	N	Resolution	[23]
Trabelsi, 2015	Tunisia	36 M	Y	N	Y	Y	Polytraumatism	Y	N	Fluco + AMB	N	N	Death	[23]
Trabelsi, 2015	Tunisia	18 M	Y	N	N	N	Polytraumatism	Y	N	AMB (8 d)	N	N	Resolution	[23]
Trabelsi, 2015	Tunisia	51 M	Y	N	N	N	Polytraumatism	Y	Y	Fluco (14 d) + AMB (6 d)	N	N	Resolution	[23]
Trabelsi, 2015	Tunisia	46 M	Y	N	Y	Y	Polytraumatism	Y	N	AMB (14 d)	N	Pneumonia	Resolution	[23]
Trabelsi, 2015	Tunisia	62 F	Y	N	Y	N	Acute pancreatitis	N	Y	AMB (2 d)	N	N	Death	[23]
Trabelsi, 2015	Tunisia	20 M	Y	N	Y	N	Polytraumatism	Y	N	Fluco (25 d)	N	N	Death	[23]
Trabelsi, 2015	Tunisia	73 M	Y	N	N	N	Renal failure, DM, myocardial infarction	Y	Y	None	N	N	Death	[23]
Trabelsi, 2015	Tunisia	74 M	Y	N	N	N	DM, polytraumatism	Y	Y	Fluco (13 d) + AMB (2 d)	N	N	Death	[23]
Trabelsi, 2015	Tunisia	14 F	Y	N	Y	N	Status epilepticus	Y	N	Fluco (16 d)	N	N	Resolution	[23]
Trabelsi, 2015	Tunisia	57 M	Y	N	N	N	DM, polytraumatism	Y	N	None	N	N	Death	[23]
Trabelsi, 2015	Tunisia	35 F	Y	N	N	Y	Post-operative shock	Y	N	Fluco	N	N	Resolution	[23]
Trabelsi, 2015	Tunisia	16 M	Y	N	N	Y	Polytraumatism	Y	N	Fluco (11 d) + AMB (16 d)	N	N	Resolution	[23]
Trabelsi, 2015	Tunisia	58 F	Y	N	Y	Y	Heart failure, DM	Y	N	Fluco + AMB	N	N	Death	[23]
Trabelsi, 2015	Tunisia	50 M	Y	N	Y	N	Guillain Barrè, COPD	Y	Y	None	N	N	Death	[23]
Trabelsi, 2015	Tunisia	36 F	Y	N	N	N	Polytraumatism	N	N	AMB	N	N	Resolution	[23]
Trabelsi, 2015	Tunisia	45 F	Y	N	Y	N	N	Y	N	AMB (6 d)	N	Pneumonia	Death	[23]
Trabelsi, 2015	Tunisia	21 M	Y	N	N	N	Polytraumatism	N	N	AMB	N	N	Resolution	[23]
Trabelsi, 2015	Tunisia	61 M	Y	N	Y	N	Polytraumatism	Y	N	AMB	N	N	Resolution	[23]
Trabelsi, 2015	Tunisia	26 M	Y	N	N	N	Polytraumatism	Y	N	None	N	N	Resolution	[23]
Trabelsi, 2015	Tunisia	78 M	Y	N	N	Y	Polytraumatism	N	Y	None	Splenic infarct	N	Death	[23]
Trabelsi, 2015	Tunisia	27 M	Y	N	N	N	Polytraumatism	Y	N	AMB (14 d)	N	N	Resolution	[23]
Trabelsi, 2015	Tunisia	83 M	Y	N	Y	N	Heart failure, trauma	N	N	None	N	N	Death	[23]
Trabelsi, 2015	Tunisia	42 M	Y	N	N	N	DM, polytraumatism, pneumothorax	Y	N	AMB	N	Pneumonia	Resolution	[23]
Trabelsi, 2015	Tunisia	30 M	Y	N	Y	N	DM, polytraumatism	Y	N	None	N	N	Resolution	[23]
Trabelsi, 2015	Tunisia	45 M	Y	N	N	N	DM, polytraumatism	N	N	AMB + fluco	N	Pneumonia	Death	[23]
Trabelsi, 2015	Tunisia	50 M	N	Y (pharyngeal cancer)	N	N	Pharyngeal cancer	N	N	AMB (2 d)	N	N	Death	[23]
Trabelsi, 2015	Tunisia	61 M	Y	N	Y	Y	DM, colitis	N	Y	Fluco (7 d)	N	N	Resolution	[23]
Trabelsi, 2015	Tunisia	52 M	Y	N	Y	Y	Rectocolitis	N	N	AMB (1 d)	Thrombophlebitis	N	Death	[23]
Trabelsi, 2015	Tunisia	36 F	Y	N	N	N	DM, CKD	Y	Y	None	N	N	Resolution	[23]
Trabelsi, 2015	Tunisia	25 F	Y	N	Y	N	DM, epilepsy, caustic oesophagitis	Y	N	AMB (4 d)	N	Pneumonia	Death	[23]
Trabelsi, 2015	Tunisia	38 M	Y	N	Y	N	Polytraumatism, DM	Y	N	AMB	N	N	Death	[23]
Trabelsi, 2015	Tunisia	29 M	Y	N	Y	Y	Abdominal trauma	N	Y	AMB + fluco	N	Pneumonia	Death	[23]
Trabelsi, 2015	Tunisia	30 M	Y	N	N	N	Thoracic trauma	Y	N	None	N	Lung abscess	Death	[23]
Trabelsi, 2015	Tunisia	46 M	Y	N	N	N	DM, COPD	N	Y	Fluco	N	N	Death	[23]
Trabelsi, 2015	Tunisia	32 M	Y	N	N	N	N	Y	Y	AMB	N	Pneumonia	NR	[23]
Trabelsi, 2015	Tunisia	34 M	Y	N	N	N	Polytraumatism	N	N	Fluco	N	N	Resolution	[23]
Trabelsi, 2015	Tunisia	64 M	Y	N	N	N	Haemorrhagic stroke	Y	N	AMB + fluco	N	Pneumonia	Death	[23]
Trabelsi, 2015	Tunisia	60 M	Y	N	N	N	DM, pulmonary oedema	Y	N	Fluco	N	N	Resolution	[23]
Trabelsi, 2015	Tunisia	4 M	Y	N	N	N	Caustic oesophagitis	Y	Y	None	N	N	Death	[23]
Trabelsi, 2015	Tunisia	40 M	Y	N	N	N	Polytraumatism	Y	N	None	N	N	Death	[23]
Trabelsi, 2015	Tunisia	18 M	Y	N	N	N	Polytraumatism	Y	N	AMB	N	Pneumonia	Resolution	[23]

ALL: Acute lymphoblastic leukemia; AMB: amphotericin B; AML: acute myelogeNus leukemia; BMT: bone marrow transplantation; BSI: bloodstream infection; CKD: chronic kidney disease; CMV: cytomegalovirus; COPD: chronic obstructive pulmonary disease; DM: diabetes mellitus; MIC: minimal inhibitory concentration; D: days; Fluco: fluconazole; Y: Yes; N: No; NR: Not reported; PN: parenteral nutrition; TB: tubercular; UTI: urinary tract infection; * < 6 months; ^: in years when not specified; **: MICs of amphotericin B was ≤1 μg/mL for 94.5 % of strains; MICs of fluconazole was <8 μg/mL for 87.2 % of strains.

## Data Availability

Not applicable.

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
