# Peer review of "Candida lipolytica Bloodstream Infection in an Adult Patient with COVID-19 and Alcohol Use Disorder: A Unique Case and a Systematic Review of the Literature"

_antibiotics, 2023, doi:10.3390/antibiotics12040691_

Round 1

Reviewer 1 Report

The manuscript “Candida lipolytica bloodstream infection in a COVID-19 adult patient with alcohol use disorder: a unique case and a systematic review of literature” has been reviewed. The authors reported a case of bloodstream infection by C. lipolytica in a COVID-19 adult patient with alcohol use disorder and performed a systematic review. The manuscript is well written and the subject matter is extremely relevant, but it needs minor modifications. Please find below my remarks:

L43 – Please italicize “albicans”

L50 – Please replace “flora” with “microbiota”

Table 1 – Please resize the table to improve the presentation and visualization of the data

Introduction - I suggest including a short paragraph discussing C. lipolytica virulence factors and the pathogenesis of C. lipolytica bloodstream infection.

Reviewer 2 Report

The authors provided probably the first case report of C. lipolytica bloodstream infection in a man with an alcohol withdrawal syndrome and mild COVID-19. Moreover, the authors prepared a systematic review according to PRISMA statements and found 89  cases of C. lipolytica bloodstream infections. My comments and suggestions are below:

1. In the previous review article about Y. lipolytica and its infection cases (doi: https://doi.org/10.1007/s11274-018-2583-8), the authors described also other than bloodstream infections. Moreover, the authors of the reviewed article omitted 2 articles, but they are difficult to access, and they are in Spanish/French, so they may have been omitted in this case.

2. The table caption should be changed to show that the cases are connected with the blood cultures.

3. Moreover, the current appearance of the table makes it difficult to analyze the results.

4. All the abbreviations used in the table should be defined in the table footer.

5. The description of the table may be more expanded.

6. Have MIC/MFC values been determined for selected antifungals against this isolate?

Round 2

Reviewer 2 Report

The manuscript has been revised and the reviewer's comments were addressed.